# Biotechnological Approaches to Optimize the Production of Amaryllidaceae Alkaloids

**DOI:** 10.3390/biom12070893

**Published:** 2022-06-25

**Authors:** Manoj Koirala, Vahid Karimzadegan, Nuwan Sameera Liyanage, Natacha Mérindol, Isabel Desgagné-Penix

**Affiliations:** 1Department of Chemistry, Biochemistry and Physics, Université du Québec à Trois-Rivières, Trois-Rivières, QC G9A 5H7, Canada; manoj.koirala@uqtr.ca (M.K.); vahid.karimzadegan@uqtr.ca (V.K.); nuwan.sameera.liyanage@uqtr.ca (N.S.L.); natacha.merindol@uqtr.ca (N.M.); 2Groupe de Recherche en Biologie Végétale, Université du Québec à Trois-Rivières, Trois-Rivières, QC G9A 5H7, Canada

**Keywords:** amaryllidaceae alkaloids, bioactive molecules, biotechnological approach, biosynthesis, in vitro cultures, synthetic biology

## Abstract

Amaryllidaceae alkaloids (AAs) are plant specialized metabolites with therapeutic properties exclusively produced by the Amaryllidaceae plant family. The two most studied representatives of the family are galanthamine, an acetylcholinesterase inhibitor used as a treatment of Alzheimer’s disease, and lycorine, displaying potent in vitro and in vivo cytotoxic and antiviral properties. Unfortunately, the variable level of AAs’ production *in planta* restricts most of the pharmaceutical applications. Several biotechnological alternatives, such as in vitro culture or synthetic biology, are being developed to enhance the production and fulfil the increasing demand for these AAs plant-derived drugs. In this review, current biotechnological approaches to produce different types of bioactive AAs are discussed.

## 1. Current Challenges in the Production of Amaryllidaceae Alkaloids

Amaryllidaceae alkaloids (AAs) are isoquinoline alkaloids exclusively isolated from the Amaryllidaceae plant family. AAs are structurally diverse biomolecules classified into different types (or groups) based on their structure, biogenetic origin, or chemical nature [1,2,3,4]. They can be divided into nine groups: norbelladine, cherylline, galanthamine, lycorine, homolycorine, crinine, pancratistatin, pretazettine, and montanine, according to their ring type and biosynthetic origin [5]. Among all the AAs, the reversible acetylcholinesterase inhibitor galanthamine is yet the only one to be approved for medicinal purposes to treat early symptoms of Alzheimer’s disease in humans [6]. Although the mechanism of action is still not fully understood, two main speculations have been proposed. Galanthamine reversibly, competitively, and selectively inhibits acetylcholinesterase, an enzyme known for acetylcholine degradation, so that the neurotransmitter associated with memory formation and learning will be available for a longer time in the synaptic cleft of cholinergic neurons to transfer neuro-signals [7,8]. In addition, galanthamine allosterically binds to nicotinic acetylcholine receptors of the central nervous system that control the release of different types of neurotransmitters, altering their conformation leading to an increase in neurotransmitters secretion [7]. AChEs inhibitory action of galanthamine also decreases the level of reactive oxygen species [9], oxidative stress being a common adverse effect of many human diseases such as Alzheimer’s, Parkinson’s, Down syndrome, cancer, etc., which hints towards a neuro-protective effect.

Up to now, galanthamine production has mainly relied on natural resource exploitation from species such as *Galanthus*, *Leucojum*, *Narcissus*, etc. Providing galanthamine to the 55 million people living with dementia cannot solely rely on plant source, and in the case of some species like *Leucojum*, it has already endangered biodiversity of wild population in the past years [10]. As an alternative strategy, chemical synthesis of galanthamine has been attempted [11,12]. However, multi-step synthesis of structurally complex compounds such as galanthamine is not economically competitive compared to extraction from native plants due to the low final yield [13].

Lycorine, another prominent AA, exhibits a broad spectrum of biological activities, including anti-viral, anti-bacterial, anti-parasitic, and anti-inflammatory properties, and it has been particularly studied for its anticancer activity [14]. Lycorine’s antitumor potency involves several pathways, such as induction of apoptosis and necrotic cell death, inhibition of cell cycle, of autophagy, and of metastasis, probably aiming at multiple molecular targets [14]. Its high cytotoxic potency at low concentrations makes lycorine’s structure an interesting leading molecule for the design of new anticancer drugs. Recently, the less abundant AA cherylline was also shown to possess antiflaviviral potential, inhibiting both dengue and Zika viruses at the viral RNA replication step, with EC_50_ of 8.8 µM and 20.3 µM, respectively [15]. In fact, novel AAs with anti-acetylcholinesterase, anti-viral, cytotoxic, anticonvulsant, antitumor, hypotensive, and anti-inflammatory properties are continuously discovered [3,5,15,16]. Their pharmacological potential depends on their complex chemical structure, including their region-specific functionalization and chirality [17]. Consequently, it is often challenging and not always cost-effective nor ecological to chemically synthetize intact structures of AAs. Although there are some reports on successful chemical synthesis of AAs such as galanthamine, lycorine, or cherylline, the multiple steps involved lead to a low overall yield [18,19,20,21,22,23,24].

Currently, plants are the main source of AAs. Even though many AAs with interesting pharmacological potentials were identified, clinical application and further research are restricted, mainly because of the variable and low production levels *in planta*. For example, cherylline-type AAs are rare because they are specifically recovered from few species of *Crinum* (0.004% crude alkaline solution in *C. powelli*) and in about twenty ornamental cultivars of *Narcissus* (extraction from the leaves of jonquilla and apodanthus daffodil cultivar “sundial”) [25,26]. Furthermore, the synthesis and accumulation of AAs in plants vary with environmental and seasonal changes throughout the year. For example, the alkaloid content of *Cyrtanthus contractus* changes from 667.4 to 1020.6 μg/g between months of the same year [27]. In addition, massive harvesting for the extraction of alkaloids decreases the number of Amaryllidaceae in nature and have led some species to become endangered, such as *Narcissus asturiensis* [28].

Therefore, intensive research on sustainable alternative techniques is carried out to achieve economical and eco-responsible production of pharmacologically active AAs [29,30,31,32]. In recent years, a book chapter by Laurrain-Mattar et al. and two reviews well describing such techniques were published, emphasizing the growing interest on this matter [33,34,35]. As an alternative to native source harvesting or chemical synthesis, biotechnological strategies offer many advantages to sustainably produce AAs (Figure 1). This includes cultivation of plant and plant parts in artificial systems, or synthetic biology (metabolic engineering) of heterologous host for AA production (Figure 1). In addition, different approaches have been reported to optimize AAs yield in plants and in vitro cultures. In this review, we discuss recent progress on the biotechnological approaches and overall factors affecting their efficiency, together with future perspectives to boost the synthesis and accumulation of AAs.

## 2. In Vitro Techniques to Produce Amaryllidaceae Alkaloids

In vitro systems hold the beneficial ability to continuously produce plant specialized metabolites in a sustainable way. They also enable the control of environmental factors, a complicated task in nature, providing the opportunity to analyze the effect of different variables in the production of specialized metabolites [36]. Several therapeutic and marketed metabolites have been produced using in vitro cultures, such as the anti-bacterial and anti-inflammatory naphtoquinone shikonin from *Lithospermum erythrorhizon*, the chemotherapeutic agent paclitaxel from *Taxus baccata,* the antioxidant saponins from *Panax ginseng* cells, the bioactive alkaloids berberine and sanguinarine from *Coptis japonica* and Papaver somniferum cultures, respectively, as reviewed in [37] and [38]. Thus, reports on many other plant families have shown that in vitro cell culture techniques can be a fantastic platform to produce specialized molecules and to understand their biosynthesis [5,39]. The specific interest in Amaryllidaceae in vitro cultures as a mean to produce alkaloids was first reported in 1963 by Fales et al. [40], and has since been intensively and continuously studied.

In vitro techniques involve the transfer of healthy sterile explants into artificial condition using suitable growth media. It can be applied to grow whole plants, plant parts, or undifferentiated tissues. Micropropagation, a technique that enables rapid vegetative/clonal multiplication of plants from limited or small size plants, has been successfully applied with Amaryllidaceae species such as *Rhodophiala pratensis*, *Lapiedra martinezii*, *Eucrosia stricklandii*, and *Lycoris sprengeri*, leading to plant development with similar morphometric traits [28,41,42,43,44]. Other cultivation methods of plant material in vitro (bulblets, seedlings, plantlets, shoots, roots, shoot–clump, callus) also provide an interesting opportunity to produce AAs, being effective for both conservation, long term growth, and industrial purpose. Specifically, callus induction is defined as the growth of undifferentiated tissues from any plant parts. Because different plant parts produce different amounts and types of alkaloids, the obtained type of callus and its metabolite content may be related to the type of tissue used as a starting material [45]. The production of uncommon but interesting AAs such as cherylline, tazettine, haemanthamine, mesembrenone was reported in various studies of in vitro propagation of Amaryllidaceae species (Table 1). For example, up to 6.9 mg/100 g DW of anti-acetylcholinesterase and anti-viral cherylline [15,46] was observed in bulblets cultures of *Crinum moorei* cultivated in presence of charcoal [47]. Bulbs are known to accumulate high concentrations of different types of alkaloids. Many studies on in vitro culture of Amaryllidaceae plant used the twin scale size of inner part of bulbs as starting material because the inner part of the bulb contains more meristem tissue than the outer part, and there is less chance of contamination. Tissues generated from in vitro culture of Amaryllidaceae plant show different range of alkaloid content depending upon cell differentiation (Table 1 and Table 2). In general, the more differentiated tissues (such as bulblet) produce higher alkaloid content as compared to undifferentiated tissue (callus) (Table 1). Therefore, the alkaloid production from in vitro system have generally focused on differentiated tissues.

Additionally, the production of well-known bioactive AAs has been investigated in in vitro cultures (Table 2). Although undifferentiated tissues do not always yield to high amounts of alkaloid, elicitation helps increase the yield of various AAs in in vitro cultures (Table 2) [50,51]. Still, this technique is advantageous because it can maintain growth for long periods of time, and callus can be used as a gateway for micropropagation, plant cell suspension cultures, or other in vitro systems to produce alkaloids. Indeed, because of somaclonal variations, shoots grown from callus displayed increased galanthamine production in some studies [50].

**Table 2 biomolecules-12-00893-t002:** Production of Amaryllidaceae alkaloids in in vitro cultures following elictor treatment.

Species	Culture Condition (Tissue Type)	Amaryllidaceae Alkaloid	Yield and Type of Condition	Elicitor and Yield	Ref.
*Narcissus confuses*	Liquid-shaken culture (shoot clumps)	Galanthamine	ut. 2–2.5 mg / culture	MJ (3.8 X)	[52]
*N. pseudonarcissus* cv. carlton	Callus	Galanthamine	ut. 7.88 μg/g FW	MJ (5.6 X) Chitosan (3 X)	[53]
*Lycoris longituba*	Liquid medium (seedling)	Galanthamine	ut. n.a.	MJ (2.71 X)	[54]
		Lycorine	ut. n.a.	MJ (2.01 X)	
		Lycoramine	ut.n.a.	MJ (2.85 X)	
	Seedling (culture in tray)	Galanthamine	white light n.a.	Blue light (2.45 X)	[55]
		Lycorine	white light n.a.	Blue light (1.74 X)	
		Lycoramine	white light n.a.	Blue light (1.92 X)	
*Lycoris chinensis*	seedling	Galanthamine	ut. n.a.	MJ (1.49 X) YE (1.62 X) SNP (1.72 X)	[56]
		Lycorine	ut. n.a.	MJ (1.37 X) YE (1.38 X)	
*Leucojum aestivum*	In vitro plants	Galanthamine	ut. n.a.	Melatonin (58.6 X)	[57]
		Lycorine	ut. n.a.	Melatonin (1.5 X)	
	Liquid shoot culture	Galanthamine	ut. n.a.	JA (1.36 X)	[58]
		Lycorine	ut. n.a.	JA (1.40–1.67 X) MJ (1.3 X)	
		Norgalanthamine	ut. n.a.	JA (2 X) MJ (2 X)	
	temporary immersion system (bulblets, leaves)	Galanthamine	ut. 372.2–1719.6 μg/g DW	MJ (468.6–2202.5 μg/g DW)	[59]
*L. aestivum* L.	RITA Bioreactor	Galanthamine	ut. n.a.	MJ (0.1 mg /g DW) ACC (0.10 mg/ g DW)	[60]
		Lycorine	ut. 0.2–0.25 mg /g DW	MJ (0.6 mg /g DW) SA (1 mg /g DW) Ethephon (0.46 mg /g DW)	
*L. aestivum* Gravety Giant	RITA Bioreactor	Galanthamine	ut. 0.08–0.1 mg/g DW	MJ (0.4 mg/g 5 X DW) SA (8 X) ACC (0.60 mg/g)	[60]
		Lycorine	ut. 0.15–0.62 mg/g DW	MJ (1.15 mg/g DW 1.85 X) SA (5 X) ACC (0.54mg/g DW, 3.6 X)	

Abbreviations; n.a.: not available; ut: untreated, basal condition, MJ: methyl jasmonate, JA: jasmonic acid; SA: salicylic acid; ACC: 1-aminocyclopropane-1-carboxylic acid; FW: Fresh weight; DW: Dry weight; X: fold change; YE: yeast elicitor; SNP: sodium nitroprusside; Ref: reference.

In addition to a proper selection of tissue and in vitro propagation technique, many other factors affect the growth and the efficiency of alkaloid production. In natural conditions, the growth of Amaryllidaceae plants and their ability to synthesize different types of alkaloids vary throughout the year and are influenced by biotic and abiotic environmental factors that affect the synthesis and accumulation of AAs [27,56,61]. Therefore, understanding and controlling both the effects of growth conditions and the plant defense response mechanisms would be helpful to increase biomass production with higher yields of specialized metabolites such as AAs. Remarkably, these factors can be monitored and optimized using in vitro methods, Studies have shown that Amaryllidaceae plants grow differently under distinctive artificial conditions. The overall goal of all culture techniques is to provide optimal growth conditions and boost the production of alkaloids. The main focus of the majority of the published research was the production of galanthamine, while there is also abundant data on lycorine production optimization. Table 2 illustrates the impact of different conditions of elicitation on AAs production in in vitro systems. Bergoñón et al. achieved a total production of 2.50 mg of galanthamine per culture by cultivating shoot–clumps in shaking-liquid media, which is the highest amount of in vitro production of galanthamine ever reported to date [62].

In the following section, we briefly discuss the effects of the different physical and chemical parameters used in the studies summarized here which may be applied on the in vitro system and eventually affect growth and development of Amaryllidaceae plant tissue culture, as well as the synthesis and accumulation of alkaloids.

### 2.1. Physical Parameters

Different cultivation methods have been tested to optimize the in vitro growth of Amaryllidaceae plants. This includes solid media culture, shaken-flask submerged condition, temporary immersion, or fully-submersive techniques in RITA^®^ bioreactor (Table 1 and Table 2) [43,62,63]. Pavlov et al. grew *Leucojum aestivum* 80 shoot culture in shaken-flasks following induction from callus and monitored their growth-index. They observed that the maximum biomass was obtained at day 35 and that AAs biosynthesis intensified at late exponential to early stationary growth phases [29]. In a following study, to optimize production of target metabolites (mostly galanthamine), *L. aestivum* 80 shoots were cultivated in temporary immersion RITA system with a higher growth index (i.e., 2.98) compared to shaken-flasks culture. The main advantage of the temporary immersion system was that cultivated shoots increased significantly, while shoot was generated from meristematic cell [63]. The system was further improved for *L. aestivum* 80 shoot culture using advanced modified gas column bioreactor with a 1.7 mg/L maximum production of galanthamine [64]. *L. aestivum* shoot cultures show balanced growth at all tested regimes in this bubble-column bioreactor. Similar techniques using twin scale from bulbs as starting materials have shown that *Narcissus confuses* shoot–clump culture in liquid-shake medium lead to an efficient micropropagation system to produce galanthamine (2.50 mg per culture in day long photoperiod) [62].

In addition to the type of in vitro cultivation system, temperature is a key factor that modulates both growth and alkaloid production in Amaryllidaceae species. Some studies have shown that among different culture temperature (i.e., 18 °C, 22 °C, 26 °C, and 30 °C), the maximum yield of galanthamine was achieved at 26 °C, whereas the best combination of highest amount of dry mass (20.8 g/L) and galanthamine content (1.7 g/L) was achieved at 22 °C when shoot culture were grown under 18 L/(L·h) flow rate of inlet air [63,64,65]. Others have been more successful at lower temperatures. Ivanov et al. obtained 18 different AAs from shoot culture of *L. aestivum* 80 and reported that lower temperature (18 ℃) favored the production of galanthamine, while inhibiting the production of lycorine- and haemanthamine-types of alkaloids. They concluded that temperature possibly alters the activity of the enzymes, catalyzing pheno-oxidative coupling reaction of 4′-*O* methylnorbelladine [65].

Light is another important factor that can boost the production of AAs in in vitro cultures. In general, studies suggest that light has a positive impact on alkaloid production in Amaryllidaceae tissues [29,31,62,66]. In shoot cultures of *N. confuses*, light affected both morphology and alkaloid content [62]. In *N. tazetta* L., bulblets and leaves proliferation per explant was higher in light condition light/dark photoperiod (16/8 h), as compared to a 24 h dark condition. Additionally, regenerated bulblets contained 40 µg/g dry weight of galanthamine under exposed photoperiod compared to 20 µg/g dry weight for 24 h dark condition [66]. Not only the light condition, but also its quality have been studied in relation to the production of AAs. For example, it was observed that increases in the production of galanthamine (2.45 times), lycorine (1.74 times), and lycoramine (1.92 times) were generated by blue light condition compared to white light in in vitro plantlets of *Lycoris longituba* [55]. Altogether, these studies demonstrate that a complex combination of physical parameters impact alkaloid production, and that cultivation system, light, and temperature should be optimized in each system, and may vary in between species.

### 2.2. Chemical Factors

Plants require diverse types of micro- and macro-elements for their growth and development. Generally, the type of media and the addition of growth factors, hormones, or other regulators such as charcoal affect both the growth and the production of metabolites [67]. Activated charcoal is known to promote plant cells and callus differentiation, possibly through the induction of genes from the phenylpropanoid biosynthesis pathway, which could lead to an increase alkaloid production [47,67,68]. The quantity of carbon and nitrogen in the media affects both the growth and the production of specialized metabolites. Studies have shown that the type and the concentration of different carbon source play an important role in plant tissue culture. In plant tissue culture research, sucrose is widely used as a carbon source. The effect of its concentration on *N. confuses* culture was measured following growth in liquid-shaked medium by Sellés et al. (1997) [49]. They showed that sucrose concentration affects biomass production, as well as galanthamine synthesis. Assessing the effects of concentrations ranging between 3%—18% of sucrose, optimal combination of growth and galanthamine production was achieved with 9% [49]. In 2020, Ptak et al. showed that both the concentration and the type of carbohydrate are critical for synthesis of AAs. Similar to Sellés study, they demonstrate that the highest amount of *L. aestivum* biomass was obtained with 9% sucrose when cultivated in RITA^®^ bioreactor. However, they also show that the use of other types of sugar can increase the success of AAs synthesis, as the highest amount of galanthamine was recorded with 3% fructose [69].

The type of phytohormones and their concentration play a key role in tissue survival and differentiation, including in organogenesis and eventually in the synthesis of AAs. Auxins, abscisic acid, cytokinins, ethylene, and gibberellins are commonly recognized as the five main classes of naturally occurring plant hormones. The isolated or combined effect of auxin or cytokinin on in vitro cultures’ nutrient uptake and metabolism was demonstrated [70]. In 2011, Tahchy et al. observed that an increment of auxin (2,4-dicholophenoxyacetic acid or 2,4-D) in growth media reduced the survival of in vitro cultured tissues of *N. pseudonarcissus, Galanthus elwesii*, and *L. aestivum*, whereas an increment of cytokinin (6-benzylaminopurine, BAP) increased it [71]. The concentration and the type of phytohormones also influence the type of tissue that will develop, which eventually affects the alkaloid profile. Hence, the alkaloid profile of *L. aestivum* and *N. pseudonarcissus* cv. Carlton obtained from in vitro system revealed that differentiated cells are more suitable for production of AAs as compared to undifferentiated cells (callus), in which alkaloid contents was lower [29,50]. A study completed on different varieties of *Narcissus* showed that undifferentiated calli development was induced following treatment with auxin concentrations of 25 µM of 1-napthaleneacetic acid (NAA), 50 µM of 2,4-D, and picloram, whereas organogenesis only happened on calli treated with NAA or using higher concentration of picloram. Interestingly, AAs demethylmaritidine and tazettine were only detected on the differentiated callus [51]. Most studies have confirmed that the amount and the type of auxin correlated with the alkaloid profile and with tissue differentiation during in vitro culture [50,51], although no consensual combination can be clearly defined at this point.

Although the eco-physiological role of many plants specialized metabolites is not clear, studies demonstrated that different biotic and abiotic factors or signaling agents (elicitors) can boost the production of AAs [56,61,72,73]. Different types of elicitors such as fungal elicitors, methyl jasmonate, jasmonic acid, salicylic acid, and melatonin have been used to enhance the synthesis of AAs in in vitro culture (Table 2). The induction of AAs using methyl jasmonate treatment on seedlings of *Lycoris aurea* was well studied by Wang et al. (2017) [74]. Others have deciphered that methyl jasmonate and jasmonic acid increased the production of AAs in *L. aestivum.* shoot cultures cultivated in submerged condition by stimulating two enzymes involved in the formation of AA precursors [58]. Melatonin addition (10 µM) during in vitro culture of *L. aestivum* L. reduced the negative effect of NaCl (i.e., salt stress) and enhanced the biomass production together with an increased accumulation of galanthamine and lycorine by 58.6 and 1.5 folds, respectively (Table 2) [57]. In conclusion, the optimization of media components and of elicitor type in in vitro culture of Amaryllidaceae provides an alternative and sustainable source of AAs.

## 3. Genetic Engineering of Heterologous Host for Alternative Production of AAs

Bioengineered microbial hosts that grow rapidly can produce plant target specialized metabolites faster as compared to whole plant systems. In addition, the production of plant metabolites in heterologous hosts can reduce downstream extraction process, which eventually becomes more economically sustainable. For the successful synthesis of plant metabolites such as AAs, heterologous hosts require the introduction of reconstructed biosynthetic pathway, requiring key enzymes. This requires comprehensive knowledge of the enzymatic reactions involved in the biosynthesis of the compound of interest in the native host organisms (i.e., plants).

### 3.1. Molecular Understanding of Amaryllidaceae Alkaloids Biosynthesis

Even though the pharmacological aspect of AAs has extensively been explored, the full understanding of the AA biosynthetic pathway and the characterization of enzymes responsible for catalyzing the different biosynthetic reactions demand more efforts. This knowledge would enable the establishment of improved systems or sustainable platforms for the production of these valuable biologically active compounds. Combined application of early labeling study followed by latest omics strategies have accelerated the discovery of AAs biosynthetic enzymes [4,75]. After the proposition of the biosynthetic route of different intermediates, several biosynthetic enzymes were predicted based on the nature of the biochemical reaction and by homology with enzymes involved in alkaloid biosynthesis of other plant families. Databases generated from transcriptomic and metabolic analysis of different species of Amaryllidaceae support the presence of different enzyme families involved in AAs pathway [45,54,76,77,78,79,80,81].

The AA biosynthetic pathway utilizes two common amino acids, namely _L-_tyrosine and L-phenylalanine, as building blocks to produce a vast range of alkaloids with diverse biological activities. The first reactions of AA biosynthesis involve the formation of the ‘precursors’ from the phenylpropanoid and tyramine pathways (Figure 2). As such, _L-_tyrosine is decarboxylated by the enzyme tyrosine decarboxylase (TYDC) to yield tyramine while the production of the second building block, 3,4-dihydroxybenzaldehyde (3,4- DHBA), is achieved via the phenylpropanoid pathway by the action of enzymes such as phenylalanine ammonia-lyase (PAL), cinnamate 4-hydoxylase (C4H), *p*-coumarate 3-hydroxylase (C3H), to name but a few. TYDC was characterized from *Lycoris radiata,* a galanthamine producing Amaryllidaceae plant [82]. The functional characterization of PAL and C4H in *L. radiata* was reported using heterologous expression in bacteria [77].

Despite having a remarkable diversity in structure and biological activity, all AAs are derived from a common intermediate, norbelladine. The condensation of tyramine and 3,4-DHBA yields norbelladine and was shown to be catalyzed either by norbelladine synthase (NBS) or by noroxomaritidine/norcraugsodine reductase (NR), in both cases with low yield [77,83]. NBS was characterized from *N. pseudonarcisus* king Alfred and *L. aestivum* [77]. GFP-tagged *La*NBS and CFP-tagged NR showed that both enzymes are localized to the cytosol, which suggests that the first committed step of AA biosynthesis probably occurs in the cytosol [77,84].

Norbelladine can either be utilized directly to generate norbelladine- and cherylline-type AAs or be further methylated by norbelladine 4′-*O*-methyltransferase (N4OMT) to give 4′-*O*-methylnorbelladine (Figure 2). The structural feature of cherylline-type of AAs suggests the occurrence of 3′-*O*-methylation during the biosynthesis of these types of AAs, although it remains to be proven. The specific synthesis of both 3′-*O*-methylated and 4′-*O*-methylated AAs suggest that regioselective methylation is important to determine the types of the end product of AAs biosynthesis route. The characterization of norbelladine OMT from *Narcissus* sp. *aff. pseudonarsissus* suggests that methylation by *Np*N4OMT happens specifically at 4′-*O* position of norbelladine [78]. However, later studies on *L. radiata* OMT (*Lr*OMT) propose that methylation can occur either in the 3′-*O* or 4′-*O* position of norbelladine, 3,4-DHBA, or caffeic acid. Kinetic study of *Lr*OMT indicates that it has a higher affinity for 3,4-DHBA as substrate compared to norbelladine. The methylated forms of 3,4-DHBA (i.e., vanillin and isovanillin) could also be condensed with tyramine to generate 3′ or 4′-*O*-methylnorbelladine. However, up until now, none of the possible methylated forms of 3, 4-DHBA were tested as a substrate for NBS.

One step deeper in the AA pathway, and depending on the type of phenol-coupling reaction, the 4′-*O*-methylnorbelladine can be directed to 1) galanthamine-type through *para-ortho’*, 2) lycorine-type AAs by *ortho-para’*, and 3) crinine-type of AAs by *para-para’* phenol coupling reactions (Figure 2). These types of C-C phenol-coupling reactions are putatively catalyzed by members of the cytochrome P450 enzyme family. For example, *Np*CYP96T1 was shown to catalyze the *para-para’* oxidative reaction of 4′-*O*-methylnorbelladine into noroxomaritidine and was also shown to catalyzed formation of the *para-ortho’* phenol coupled product, *N*-demethylnarwedine, as less than 1% of the total product [85]. Aside from CYP96T1 and NR, there are no other steps (genes or enzymes) that have been identified in the formation of phenol-coupled AA-types to date (Figure 2).

Plants synthesize specialized metabolites by using complex biosynthetic routes that derive from primary metabolic pathways. AAs biosynthesis is a multifaceted process that involves different regulatory elements and gene functions. The expression of certain genes involved in plant metabolism also changes with different climatic and environmental factors [27]. Furthermore, it also varies within different developmental stage of plant [45]. It remains challenging to correlate gene expression and metabolite accumulation *in planta*, as the site of metabolite synthesis may differ from the site of accumulation. For example, nicotine biosynthesis occurs in the root of tobacco but accumulates in the aerial part of the plant [86], whereas morphine biosynthesis starts in sieve elements of the phloem but accumulates in adjacent laticifers cells in opium poppy [87]. As such, in vitro cultures have been an essential tool to decipher the alkaloid biosynthesis pathway. In 2011, Tahchy et al. used deuterium-labeled precursors fed to in vitro cultures of *L. aestivum*. In this study, the authors followed the transfer of labeled precursor 4′-*O*-methyl-d_3_-norbelladine from media into shoot and then its metabolization into lycorine and galanthamine. This study demonstrated that 4′-*O*-methylated-norbelladine was a key intermediate AAs [88]. Until now, AAs specific genes such as *NBS, N4OMT, CYP96T1,* and *NR* have been characterized and confirmed from *Leucojum* sp., *Narcissus* sp., *Lycoris* sp. cultures [44,45,46,47], however, our molecular understanding regarding this complex biosynthesis route of AAs and its regulation is still unclear. Furthermore, the pattern of relative expression of putative AAs biosynthetic genes (in fields versus in vitro and in differentiated versus undifferentiated tissues of *Narcissus* development) added some clear knowledge regarding their role in alkaloid biosynthesis [89]. A study performed on callus culture of *L. radiata* showed how different factors, such as temperature (cold treatment), osmotic pressure (PEG treatment), or elicitor treatment (methyl jasmonate), can influence *Lr*OMT gene expression pattern [90]. Thus, in vitro system cultures are a powerful tool to uncover AAs biosynthesis and gene regulation that should be thoroughly exploited.

### 3.2. Synthetic Biology for AA Biosynthesis

Although the complete biosynthetic pathway of AAs is not resolved, and up to now the AAs demand has not been sufficiently fulfilled by a plant source, a synthetic biological approach could be a powerful approach to produce AAs. Recent achievements in synthetic biological approaches include the production of complex biomolecules such as noscapine (a benzylisoquinoline alkaloid from opium poppy) and its halogenated derivatives (anticancer) in *Saccharomyces cerevisiae*, assembling 30 biosynthetic enzymes from plant, bacteria, and mammal, with yeast itself including seven plant endoplasmic reticulum localized genes [91]. This success gives hope for producing complex biomolecules such as AAs by using a synthetic biological approach.

Proper selection of host organism is the starting point of synthetic biological approach. The chosen organism should be producing (or easily modified to produce) enough core metabolites such as aromatic amino acids, _L-_phenylalanine, and _L-_tyrosine, precursors needed for the biosynthesis of target specialized metabolite such as AAs. Selection of host species will also rely on prior knowledge of their ease of engineering, established cloning tools, culture techniques, and possibly scaling up to industrial requirements. Due to rapid growth and easy handling, microbial hosts such as yeast (*Saccharomyces cerevisiae*), and to a lesser extent *Escherichia coli,* were used to produce plant-derived high value alkaloids like morphinan alkaloids [92,93,94]. Furthermore, the production of aromatic amino acid (precursor for AAs) and associated upstream gene/enzyme were well studies in these hosts [95]. Precursors such as _L-_tyrosine and *p*-coumaric have been already produced in *E. coli* [96,97]. Recently, unicellular photosynthetic organisms such as microalgae and cyanobacteria became interesting research platforms because of their unicellular physiology, together with their photosynthetic, heterotrophic, and mixotrophic lifestyles [98]. Moreover, plant-based genetic engineering technique is also emerging in model plants such as *Nicotiana tabacum* and *N. benthamiana* [99].

Once a host organism is selected, availability of precursor molecules can be enhanced by modifications to its metabolic pathway, such as gene deletions, swapping of endogenous enzymes with more active homologues, or overexpression of endogenous metabolic genes. Then, a route to the desired specialized metabolites can be planned and implemented. A candidate pathway is first outlined through selection of stepwise chemical intermediates leading from host metabolism to the target compound, followed by selection of enzymes to carry out each specified reaction [100,101]. Even though the lack of knowledge in the AAs biosynthetic pathway hinders this approach as of yet, it could be partially overcome by creating libraries of gap-filling genes candidate generated from huge plant transcriptomic database, as available for thousands of plants or as part of the PhytoMetaSyn project [102]. In addition, the decrease in the cost of DNA synthesis helps accelerate gene characterization from its native source and ultimately facilitate the production of complex biomolecules like AAs [102,103,104]. Such work was done to produce polyketides. Soon, platform of synthetic approach will not only provide techniques to produce AAs but also help in the biosynthesis of novel AAs derivatives with improved biological and physiological properties. In example, once the complete identification of genes encoding enzymes required for the biosynthesis of galanthamine is achieved, one more enzyme could be added in the transgenic construct that could involve glycosylation, shifting the polarity of parent molecule, and eventually improving drug uptake by the human body.

## 4. Conclusions

Studies on the production of AAs using in vitro systems have mainly focused on the commercially available galanthamine and the abundant lycorine. Nowadays, the demand for natural therapeutic metabolites obtained from plants is growing fast, however, overexploitation of the native plants to meet this demand will be insufficient and endanger the biodiversity of wild populations. Alternative chemical synthesis requires a multi-step process to produce intact complex compounds. Fortunately, biotechnological approaches, including in vitro platforms or synthetic biology, are promising strategies to establish a more reliable, economic, and environmentally friendly system for the production of plant-derived metabolites. Currently, the production of AAs from in vitro systems does not achieve the levels produced by wild type plants. However, they have several advantages, i.e., they enable the production of target metabolites independently from environmental factors affecting the production yield, biodiversity concerns, and land usage; they facilitate the discovery of biosynthetic pathway and the understanding of its regulation in a short period of time. Currently established platforms of in vitro systems can be used to determine the effect of different variables on plants in a controlled environment with stable chemical and physical parameters. Notable effects of biotic and abiotic stresses on AA biosynthesis and accumulation in in vitro system can be used as a basis platform for transcriptomic and metabolomic level studies, which generate a huge amount of data not only regarding production of AAs but also on their regulation *in planta*. This generated data can serve as fundamental units for the synthetic biological approach. It can be utilized to: (1) to establish an in vitro production system with optimized parameters economically comparable to extraction from natural sources yet sustainable, decreasing the need for native plants harvesting; (2) be linked to other branches of science such as bioinformatics, cell biology, and biochemistry; (3) to produce metabolites in a fast-growing heterologous organism such as yeast, bacteria, and new emerging platforms like microalgae.

In this way, research combining biologists, biochemist, bioengineers, physicists, and computer scientists will enhance deep understanding on AAs metabolism and thereby enable their (re)design in selected heterologous hosts such as bacteria or yeast systems.

## Figures and Tables

**Figure 1 biomolecules-12-00893-f001:**
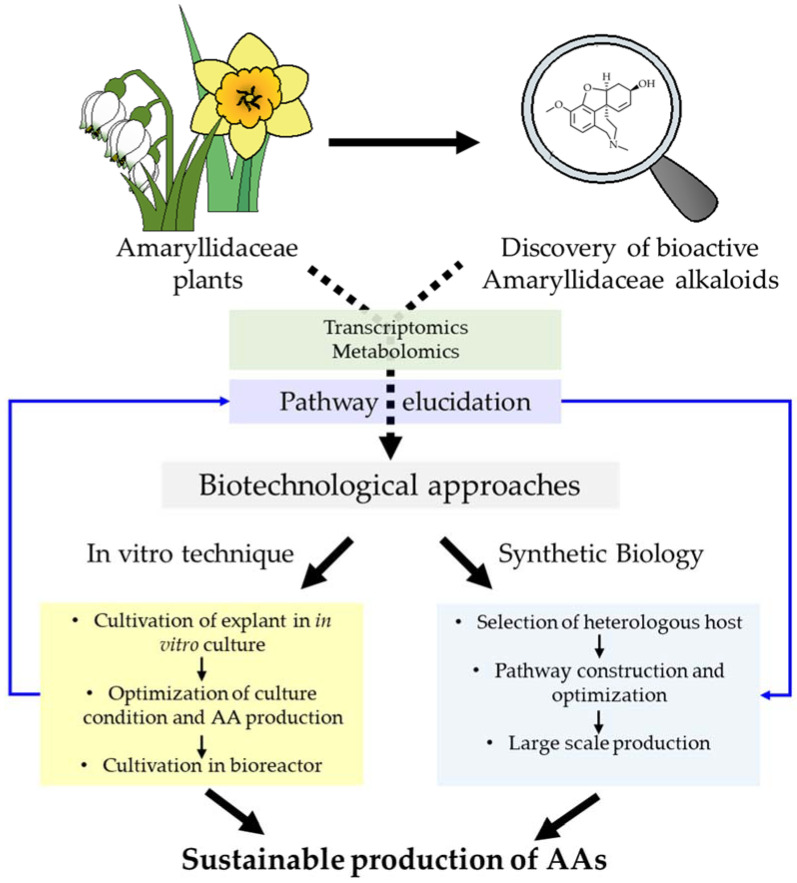
Current and future biotechnological approaches to produce Amaryllidaceae alkaloids.

**Figure 2 biomolecules-12-00893-f002:**
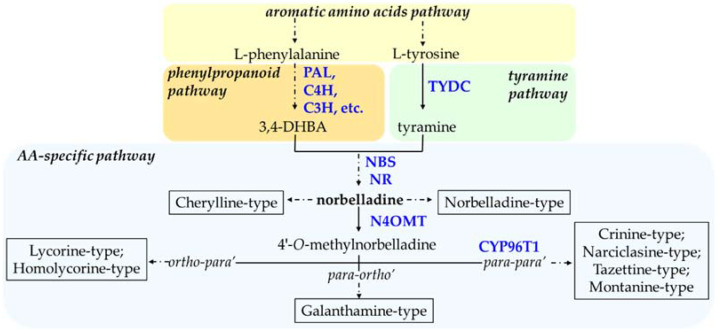
Biosynthetic routes to main types (boxed) of Amaryllidaceae alkaloid (AA). Arrows without labeling reflect chemical reactions where no enzyme was characterized. Enzymes that have been identified are labeled in blue. A solid arrow shows one enzymatic step, whereas a broken arrow symbolizes multiple enzymatic reactions. Following 4′*O*-methylnorbelladine, the regioselective *phenol-phenol*’ coupling reaction is indicated in the broken arrow, leading to various AA-types. Enzyme abbreviations: 3,4-DHBA, 3,4-dihydroxybenzaldehyde; PAL, phenylalanine ammonia-lyase; C4H, cinnamate 4-hydroxylase; C3H, coumarate 3-hydroxylase; TYDC, tyrosine decarboxylase; NBS, norbelladine synthase; NR, noroxomaritidine/norcraugsodine reductase; N4OMT, norbelladine 4′-*O*-methyltransferase; CYP96T1, cytochrome P450 monooxygenase 96T1.

**Table 1 biomolecules-12-00893-t001:** Yields of uncommon AAs of therapeutical interest in in vitro cultures.

Target Metabolites	Species	Tissue Type	Maximum Yield	Ref.
Cherylline	*Crinum moorei*	Bulblets	6.9 mg/100 g DW	[47]
Haemanthamine	*Rhodophiala pratensis*	Callus	6.9 µg/mg Ext	[41]
	*Narcissus* cv. Hawera	Plants	25.5 μg/100 mg Ext	[48]
Powelline	*Crinum moorei*	Bulblets	46.84 mg/100 g DW	[47]
Tazettine	*Rhodophiala pratensis*	Callus	2.68 µg/mg Ext	[41]
	*Narcissus confuses*	Shoot–clump culture	0.043 % DW	[49]
Mesembrenone	*Narcissus pallidulus*	Plants	337.6 μg/100 mg Ext	[48]
	*N.* cv. Hawera	Plants	214.8 μg/100 mg Ext	[48]

DW = Dry weight, Ext: Extract.

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
