# Peer review of "Biotechnological Approaches to Optimize the Production of Amaryllidaceae Alkaloids"

_biomolecules, 2022, doi:10.3390/biom12070893_

Round 1
Reviewer 1 Report
1. The authors need to select 1-2 critical AAs, for example, galanthamine and/or lycorine, and give an in-depth summary using 1-2 paragraphs for their biochemical, biotechnological, and bioactivities and future directions.
2. Conclusion: The authors need to point out more clearly the future direction about how to improve the production of AAs via biotechnology or some other ways.
3. L17: restricts most of the pharmaceutical application - restricts most of the pharmaceutical applications
4. L18: and fulfil the increasing demand of these - and fulfill the increasing demand for these
5. L20: Avoid using first-person writing throughout the manuscript.
6. L39: to chemically synthetize intact structure - to chemically synthesize intact structures
7. L48: synthesis and accumulation - the synthesis and accumulation
8. L51: for extraction - for the extraction
9. L52: have lead for some - have led some
10. L81: The specific interest on - The specific interest in
11. Table 1: The horizontal grids are missing.
12: L117: to proper selection - to a proper selection
13: L127: to provide optimal growth condition - to provide optimal growth conditions
14: L394: Current established platforms - Currently established platforms
15: L398: generate huge amount - generate a huge amount
Author Response
We wish to thank the reviewer for the comments and suggestions to improve our manuscript which are truly appreciated. See attached document.
Comments and Suggestions for Authors
- The authors need to select 1-2 critical AAs, for example, galanthamine and/or lycorine, and give an in-depth summary using 1-2 paragraphs for their biochemical, biotechnological, and bioactivities and future directions.
RESPONSE: We wish to thank the reviewer for the comments and suggestions to improve our manuscript which are truly appreciated. We answered the reviewer’s concern by adding the following paragraphs on galanthamine and lycorine in the 1. Current challenges in the production of Amaryllidaceae alkaloids section:
Line 33 to 65 : “Although the mechanism of action is still not fully understood, two main speculations have been proposed. Galanthamine reversibly, competitively, and selectively inhibits acetylcholinesterase, an enzyme known for acetylcholine degradation, so that the neurotransmitter associated with memory formation and learning, will be available for longer time in the synaptic cleft of cholinergic neurons to transfer neuro signals [7, 8]. In addition, galanthamine allosterically binds to nicotinic acetylcholine receptors of the central nervous system that control the release of different types of neurotransmitters, altering their conformation leading to an increase in neurotransmitters secretion [7]. AChEs inhibitory action of galanthamine also decreases the level of reactive oxygen species [9], oxidative stress being a common adverse effect of many human diseases such as Alzheimer’s, Parkinson’s, Down syndrome, cancer, etc., this hints towards a neuro-protective effect.
Up to now, galanthamine production mainly has relied on natural resource exploitation from species such as Galanthus, Leucojum, Narcissus, etc. Providing galanthamine to the 55 millions of people living with dementia cannot solely rely on plant source, and in the case of some species like Leucojum, it has already endangered biodiversity of wild population in the past years [10]. As an alternative strategy, chemical synthesis of galanthamine has been attempted [11, 12]. However, multi-step synthesis of structurally complex compounds such as galanthamine is not economically competitive compared to extraction from native plants due to the low final yield [13].
Lycorine, another prominent AA, exhibits a broad spectrum of biological activities, including anti-viral, anti-bacterial, anti-parasitic and anti-inflammatory properties, and it has been particularly studied for its anticancer activity [14]. Lycorine’s antitumor potency involves several pathways, such as induction of apoptosis and necrotic cell death, inhibition of cell cycle, of autophagy, and of metastasis, probably aiming at multiple molecular targets [14]. Its high cytotoxic potency at low concentrations makes lycorine’s structure an interesting leading molecule for the design of new anticancer drugs. Recently, the less abundant AA cherylline was also shown to possess antiflaviviral potential, inhibiting both dengue and Zika viruses at the viral RNA replication step, with EC50 of 8.8 µM and 20.3 µM respectively [15]. In fact, novel AAs with anti-acetylcholinesterase, anti-viral, cytotoxic, anticonvulsant, antitumor, hypotensive and anti-inflammatory properties are continuously discovered [3, 5, 15, 16]. Their pharmacological potential depends on their complex chemical structure, including their region-specific functionalization and chirality [17].”
- Conclusion: The authors need to point out more clearly the future direction about how to improve the production of AAs via biotechnology or some other ways.
RESPONSE: The reviewer is correct and the conclusion was rewritten with the following addition on future directions to improve AAs via biotechnology :
“Nowadays, the demand for natural therapeutic metabolites obtained from plants is growing fast, however, overexploitation of the native plants to meet this demand will be insufficient and endanger biodiversity of wild populations. Alternative chemical synthesis requires a multi-step process to produce intact complex compounds. Fortunately, biotechnological approaches including in vitro platforms or synthetic biology are promising strategies to establish a more reliable, economic and environmentally friendly system for the production of plant-derived metabolites. Currently, the production of AAs from in vitro systems does not achieve the levels produced by wild type plants. However, they have several advantages, i.e. they enable the production of target metabolites independently from environmental factors affecting the production yield, biodiversity concerns and land usage; they facilitate the discovery of biosynthetic pathway and the understanding of its regulation in a short period of time. Current established platforms of in vitro systems can be used to determine the effect of different variables on plants in a controlled environment with stable chemical and physical parameters. Notable effects of biotic and abiotic stresses on AA biosynthesis and accumulation in in vitro system can be used as a basis platform for transcriptomic and metabolomic level studies, which generate a huge amount of data not only regarding production of AAs but also on their regulation in planta. This generated data can serve as fundamental units for the synthetic biological approach. It can be utilized to: 1) to establish an in vitro production system with optimized parameters economically comparable to extraction from natural sources yet sustainable, decreasing the need for native plants harvesting; 2) be linked to other branches of science such as bioinformatics, cell biology, biochemistry, 3) to produce metabolites in a fast growing heterologous organism such as yeast, bacteria and new emerging platforms like microalgae.
In this way, research combining biologists, biochemist, bioengineers, physicists, and computer scientists will enhance deep understanding on AAs metabolism and thereby enable for their (re)design in selected heterologous host such as bacteria or yeast systems.”
- L17: restricts most of the pharmaceutical application - restricts most of the pharmaceutical applications
RESPONSE: The sentence has been changed to: “restricts most of the pharmaceutical applications”
- L18: and fulfil the increasing demand of these - and fulfill the increasing demand for these
RESPONSE: The sentence has been changed to: “and fulfill the increasing demand for these”
- L20: Avoid using first-person writing throughout the manuscript.
RESPONSE: We thank the reviewer for this correction, the first-person was changed to: “In this review, current biotechnological approaches to produce different types of bioactive AAs are discussed.”
- L39: to chemically synthetize intact structure - to chemically synthesize intact structures
RESPONSE: The sentence has been changed to: “to chemically synthesize intact structures”
- L48: synthesis and accumulation - the synthesis and accumulation
RESPONSE: The sentence has been changed to: “the synthesis and accumulation”
- L51: for extraction - for the extraction
RESPONSE: The sentence has been changed to: “for the extraction”
- L52: have lead for some - have led some
RESPONSE: The sentence has been changed to: “have led some”
- L81: The specific interest on - The specific interest in
RESPONSE: The sentence has been changed to: “The specific interest in”
- Table 1: The horizontal grids are missing.
RESPONSE: The table 1 has been changed accordingly.
12: L117: to proper selection - to a proper selection
RESPONSE: The sentence has been changed to: “to a proper selection”
13: L127: to provide optimal growth condition - to provide optimal growth conditions
RESPONSE: The sentence has been changed to: “to provide optimal growth conditions”
14: L394: Current established platforms - Currently established platforms
RESPONSE: The sentence has been changed to: “Currently established platforms”
15: L398: generate huge amount - generate a huge amount
RESPONSE: The sentence has been changed to: “generate a huge amount”

Reviewer 2 Report
This manuscript entitled “Biotechnological approaches to optimize the production of Amaryllidaceae alkaloids” is well-organized review regarding the biotechnological production Amaryllidaceae alkaloids. The publication of this review in Biomolecules is highly recommended, as the contribution of this to the field. Prior to publication, however, the authors may need to consider the points listed below.
· The references must be uniform because some have doi others do not.
· It also lacks a substantiated discussion in relation to biotechnological techniques. Example why medium containing activated charcoal significantly induces the production of cherylline.
· It would also be interesting to discuss between the parts of plants used (bulblets, callus..).
· In Table 2, only Lycorine and Galanthamine are reported. Why not other AAs.
· Does biotechnological conditions favor the production of galanthamine and lycorine over other AAs? if yes, why?
Author Response
We are grateful and wish to thank the reviewer for the comments and suggestions to improve our manuscript which are truly appreciated. Please see attached document.
Comments and Suggestions for Authors
This manuscript entitled “Biotechnological approaches to optimize the production of Amaryllidaceae alkaloids” is well-organized review regarding the biotechnological production Amaryllidaceae alkaloids. The publication of this review in Biomolecules is highly recommended, as the contribution of this to the field. Prior to publication, however, the authors may need to consider the points listed below.
RESPONSE: We are grateful and wish to thank the reviewer for the comments and suggestions to improve our manuscript which are truly appreciated.
- The references must be uniform because some have doi others do not.
RESPONSE: The references format was revised and corrected. For some the doi was not available. Final corrections of the references will be done manually.
- It also lacks a substantiated discussion in relation to biotechnological techniques. Example why medium containing activated charcoal significantly induces the production of cherylline.
RESPONSE: We thank the reviewer for this pertinent comment. However, to our knowledge, no other work than Fennel (2003) has demonstrated the effects of activated charcoal on specialized metabolite synthesis despite the fact that it is frequently added to tissue culture media for various reasons, including the induction of root and bulblet formation and superior growth. So of these mechanisms are not yet discovered. To answer the reviewer’s comment, we added the following sentences to section 2.2. chemical factors :
” Generally, the type of media and the addition of growth factors, hormones or other regulators such as charcoal affect both the growth and the production of metabolites [67]. Activated charcoal is known to promote plant cells and callus differentiation, possibly through the induction of genes from the phenylpropanoid biosynthesis pathway, which could lead to an increase alkaloid production [47,67,68].”
- It would also be interesting to discuss between the parts of plants used (bulblets, callus..).
RESPONSE: We agree with the reviewer’s suggestions and modified the following sentences to emphasize the relevance of the choice of plant parts in the section 2. In vitro techniques to produce Amaryllidaceae alkaloids, second paragraph :
“Because different plant parts produce different amount and types of alkaloids, the obtained type of callus and its metabolite content may be related to the type of tissue used as a starting material [45].”
Added to the end of second paragraph : ‘’Bulbs are known to accumulate high concentrations of different types of alkaloid. Many studies on in vitro culture of Amaryllidaceae plant used the twin scale size of inner part of bulbs as starting material because the inner part of the bulb contains more meristem tissue than the outer part, and there is less chance contamination. Tissues generated from in vitro culture of Amaryllidaceae plant show different range of alkaloid content depending upon cell differentiation (Table 1, 2). In general, the more differentiated tissues (such as bulblet) produce higher alkaloid content as compared to undifferentiated tissue (callus) (Table 1). Therefore, the alkaloid production from in vitro system have generally focused on differentiated tissues.’’
- In Table 2, only Lycorine and Galanthamine are reported. Why not other AAs.
RESPONSE: The other few AAs reported in in vitro cultures are mentioned in table 1. Far less studies were performed on other alkaloids.
- Does biotechnological conditions favor the production of galanthamine and lycorine over other AAs? if yes, why?
RESPONSE: Most of the published experiments have focused on the production of galanthamine, probably because it is the only commercialized AAs and has economical value. In the case of lycorine, it is detectable in many species of Amaryllidaceae plants and is also reported in different conditions of in vitro culture. To acknowledge these limitations, we added the following sentence to the conclusion: “Studies on the production of AAs using in vitro systems have mainly focused on the commercially available galanthamine and the abundant lycorine.”

Reviewer 3 Report
Authors reviewed the production of Amaryllidaceae alkaloids. The manuscript is useful for researchers of Amaryllidaceae alkaloids. I think this review is worthy being published in Biomolecules after minor revision.
1. Page 3, Line 98. The maximum yield of cherylline is up to 6.9 micro g/g in the Text DW but in Table 1, 6.9 mg/100 mg DW. Which is correct?
2. Page 6, Line 176. Authors use "40 microg・g-1". Units should be unified so use "40 micro g/g".
Author Response
We wish to thank the reviewer for the comments and suggestions to improve our manuscript which are truly appreciated. Please see the attached document.
Comments and Suggestions for Authors
Authors reviewed the production of Amaryllidaceae alkaloids. The manuscript is useful for researchers of Amaryllidaceae alkaloids. I think this review is worthy being published in Biomolecules after minor revision.
RESPONSE: We wish to thank the reviewer for the comments and suggestions to improve our manuscript which are truly appreciated.
- Page 3, Line 98. The maximum yield of cherylline is up to 6.9 micro g/g in the Text DW but in Table 1, 6.9 mg/100 mg DW. Which is correct?
RESPONSE: This has been corrected to 6.9 mg/ 100 g DW in the text, which is the correct one.
- Page 6, Line 176. Authors use "40 microg・g-1". Units should be unified so use "40 micro g/g".
RESPONSE: This has been corrected to 40 µg/g and in other places too.

Round 2
Reviewer 1 Report
The manuscript has been revised accordingly.